# FEDOD: FEDERATED OUTLIER DETECTION VIA NEURAL APPROXIMATION

## ABSTRACT

Outlier detection (OD) is a crucial machine learning task with key applications in various sectors such as security, finance, and healthcare. Preserving data privacy has been increasingly important for OD due to the sensitivity of the data involved. While federated learning (FL) offers the potential to protect data privacy, it is not yet available for most classical OD algorithms, such as those based on distance and density estimation. To address this, we introduce FEDOD, the first FL-based system designed for general OD algorithms. FEDOD effectively overcomes the privacy and efficiency challenges inherent in classical OD algorithms by automatically decomposing these algorithms into a set of basic operators and approximating their behaviors using neural networks. Given the inherent compatibility of neural networks with FL, the approximated OD algorithms also become capable of privacy-preserving learning *without* data exchange. With this design, FEDOD supports over 20 popular classical OD algorithms and is readily extendable to other fields like classification and clustering. Evaluation on more than 30 benchmark and synthetic datasets demonstrates FEDOD's accuracy and efficacy over state-of-the-art baselines—compared to existing OD systems, FEDOD achieves up to $11\times$ reduction in errors and $10\times$ improvement in performance.

## 1 INTRODUCTION

Outlier detection (OD) is an important class of machine learning (ML) algorithms that identify observations or data points deviating from the expected behavior or patterns in a dataset Zhao et al. (2019), with numerous applications in security Khan et al. (2007), finance Lee et al. (2020), and healthcare Gupta et al. (2021). Among these applications, one key consideration is preserving data privacy, especially in scenarios where sensitive data is involved. For instance, OD has been used to identify rare diseases (as outliers) in patients, while cross-hospital data sharing is often prohibited due to regulatory reasons Pfitzner et al. (2021). This constraint restricts the analysis of an OD algorithm to data available in a single hospital, thereby constraining its performance.

To preserve user privacy while maintaining high predictive performance, *federated learning* (FL) Konečnỳ et al. (2016) enables a new ML training paradigm, where an ML model is trained locally on a decentralized network of agents (e.g., hospitals in our motivating example). Instead of directly sending training data to a centralized server, each agent trains a model locally using private training samples. Once the local training process is completed, agents synchronize local gradients to update the model for future training. The process is repeated until achieving the desired accuracy.

FL has been widely deployed in distributed DNN training Konečnỳ et al. (2016) and ensemble learning Smith et al. (2017). For instance, neural networks (NN; also known as "deep" methods) can conveniently learn in a distributed manner using batches, necessitating only model parameter updates without the need for data exchange Goodfellow et al. (2016). This contrasts with non-neural-network-based (also known as "classical") ML algorithms such as $k$ nearest neighbors ($k$NN) and clustering Wu et al. (2008), which are thus not directly compatible with FL.

While neural-network-based OD algorithms can directly leverage FL paradigms by performing neural network computation using *local* samples Preuveneers et al. (2018); Li et al. (2020a); Gupta et al. (2021); Nguyen et al. (2019); Pei et al. (2022); Astillo et al. (2022), most classical OD algorithms, such as $k$NN OD Angiulli & Pizzuti (2002) and local outlier factor (LOF) Breunig et al. (2000),

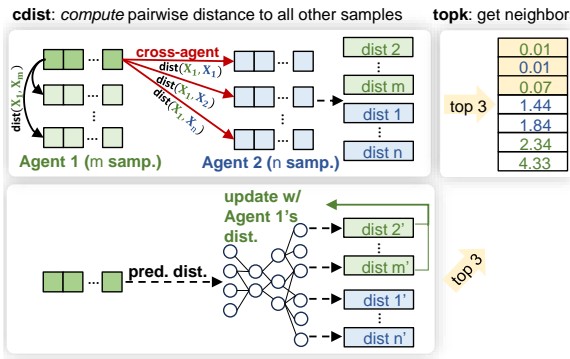

cdist: *compute* pairwise distance to all other samples    topk: get neighbors

ours: *predict* pairwise distance to all samples by NN

Figure 1: Demo. of dependency challenges in classical OD. *Top*: $k$NN OD needs to compute pairwise distances (by cdist) of a sample to *all* other samples (on both Agent 1 and Agent 2) and then find the top $k$ neighbors (by topk). Note that cross-agent distance calculation is infeasible with privacy constraints (denoted in red arrows). *Bottom*: FEDOD uses NN to approximate cdist operator to *predict* pairwise distances, where the NN is updated w.r.t. the local data on Agents 1, 2, and so on. No cross-agent dist. calc. is needed.

involve *global* inter-sample data dependencies (i.e., computing the nearest neighbors of a sample requires accessing *all* other samples), and face unique challenges when training in a federated fashion.

***Challenge 1: data dependency.*** Most classical OD algorithms have both (1) *inter-sample* data dependency, where the estimation of a sample depends on other samples , and (2) inter-feature data dependency[1], where all features contribute to the estimation of outlier scores Zhao et al. (2023). For example, $k$NN computes the $k$-th nearest neighbor of a sample as its outlier score, as shown in Fig. 1, where a larger distance indicates that the sample is far from others and thus likely to be an outlier. We need all other samples across agents to calculate and compare a sample's $k$-th nearest neighbor. This communication is not feasible under the FL assumptions where data reside on different agents, and cross-agent data sharing is strictly prohibited. In this example, the samples on Agent 1 do not have access to the samples on Agent 2 since the cross-agent distance calculation is infeasible; this prevents existing FL systems from supporting $k$NN for OD.

***Challenge 2: computational cost.*** Classical OD algorithms involve high training and inference cost, which introduces additional difficulty under the FL setting due to the limited computational power of each agent—which is often an edge device. Given $n$ training samples and $n_t$ inference samples, Appx. Table A1 shows the computational complexity for many classical OD algorithms, most of which involves a complexity higher than $O(n \cdot n_t)$. High computational complexity for inference impedes the use of classical OD algorithms in most time-critical FL applications, such as real-time credit card fraud detection Boniol et al. (2021); Jiang et al. (2022; 2023).

***Our approach.*** To address these challenges, this paper presents federated learning based outlier detection (FEDOD)[2] that can support diverse classical OD algorithms for privacy-preserving and scalable learning. Fig. 2 shows an overview of FEDOD's approach. First, we analyze a diverse group of classical OD algorithms, including distance-, density-, and tree-based algorithms, and decompose them into a small set of basic OD operators (see Fig. 2 (left) and §3). For each OD operator, we design a neural network approximation to iteratively learn the behavior of the OD operator from local data without inter-sample data dependency (see Fig. 2 (middle) and §4.1). By combining these neural networks, FEDOD approximates classical OD algorithms and leverages existing FL paradigms for privacy-preserving learning (see Fig. 2 (right) and §4.2). An example of our approach is provided in Fig. 1 bottom, where we show that FEDOD can support classical $k$NN OD by approximating distance calculation using a neural network to predict pair-wise distances other than calculating them. Note that the neural network here is trained only with regard to the local samples, without the need to access cross-agent samples. Our key contributions are:

- **The first FL system for diverse OD algorithms**. FEDOD supports more than 20 popular classical OD algorithms for privacy-preserving learning and can be easily extended to more.

- **Model decomposition and neural approximation**. FEDOD decomposes OD algorithms into basic operators and approximates them with neural networks with specialized local updating strategies, including $k$-nearest neighbors and clustering.

- **Effectiveness and scalability**. Extensive experiments on more than 30 datasets show that FEDOD outperforms the baseline with up to $11\times$ error reduction and $10\times$ speed up.

---

[1]In this work, we focus on *horizontal* FL where all agents have the same set of features and only introduce inter-feature dependency for completeness.

[2]Code is available at anonymous Google Drive: https://tinyurl.com/fedod2023

Figure 2: FEDOD overview (§2): decomposes an OD method into shared basic operators (left; §3), which are approximated by neural nets (middle; §4.1) for privacy-preserving training (right; §4.2).

# 2 FEDERATED OUTLIER DETECTION

## 2.1 PROBLEM DEFINITION

A key object of federated outlier detection is to improve the predictive performance of OD algorithms by leveraging samples across agents while preserving data privacy of each agent. Consider a horizontal FL system that supports $m$ OD models $\mathcal{M} = \{M_1, ..., M_m\}$. It can train an OD model $M \in \mathcal{M}$ using a distributed dataset located on $K$ agents, where each agent $k \in 1, ..., K$ possesses a private dataset $\mathbf{X}_k \in \mathbb{R}^{n_k \times d}$ *without* ground truth labels. Collectively, $\mathbf{X}$ is the aggregated dataset across *all* agents, where each row corresponds to an individual training sample and each column represents a feature of the sample, with a total of $d$ unified features across all agents. The FL system outputs the outlier scores, denoted as $\mathbf{O} := M(\mathbf{X}) \in \mathbb{R}^n$, which provides anomaly scores for samples across all local agents and infer on the test samples $\mathbf{O}_{\text{test}} = M(\mathbf{X}_{\text{test}})$. The key difference between FL-based OD and general OD lies in designing privacy-preserving techniques that allow agents to train the OD model $M$ collectively across all agents without data sharing, where general OD algorithms require access to all samples.

## 2.2 FEDOD'S OVERVIEW

To address the aforementioned challenges in data privacy and inference efficiency, a key idea behind FEDOD is to decompose a diversity of classical OD algorithms into a set of basic OD operators and use a neural network to approximate each operator. By doing so, existing FL frameworks designed for neural networks become available for classical OD algorithms through neural approximation.

Figure 2 provides an overview of FEDOD. Firstly, an OD algorithm is decomposed into low-level OD operators (§3), each of which is then converted into a neural network (§4.1). Finally, these neural networks are trained by FE-DOD in a federated fashion, using specialized local update strategies (§4.2). Thus, combining these neural approximations by concatenating their network architectures emulates the outputs of the OD algorithm.

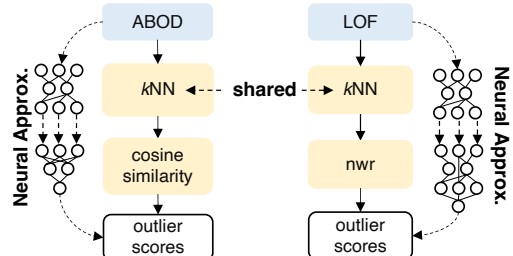

Figure 3: Example of building complex OD algorithms ABOD and LOF by *shared* basic operators, e.g., $k$NN. Each operator is approximated by a neural network (on two sides), which is trained sequentially as shown in the computational graph.

In addition to the $k$NN example in Fig. 1, Fig. 3 shows two additional examples for training ABOD and LOF in FEDOD. Specific, each complex OD algorithm is decomposed as a sequence of basic operators (e.g., ABOD is decomposed into $k$NN followed by cosine similarity), where each operator is approximated by a neural network in FEDOD. As illustrated in the figure, we use a combination of neural networks to approximate an OD algorithm; the output of one neural network becomes the input of the subsequent neural network—thus, these networks are trained together in an end-to-end fashion.

# 3 MODEL DECOMPOSITION

***Overview.*** Real-world applications often require diverse OD algorithms based on the data characteristics Zhao et al. (2021; 2022); Ma et al. (2023). To support a wide range of OD algorithms

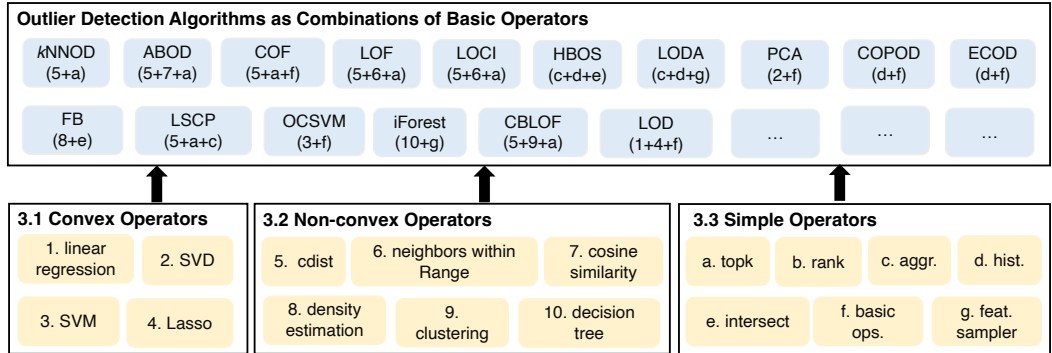

Figure 4: With model decomposition, more than 20 OD algorithms are decomposed into 4 convex operators (§3.1), 6 non-convex operators (§3.2), and 7 simple operators (§3.3). This decomposition reduces the implementation and optimization effort, and makes adding new OD operators easy.

under FL, we adapt and extend existing work on the OD decomposition model Zhao et al. (2023). The key idea is to decompose complex OD algorithms into low-level recurring operators, which are shared across diverse OD algorithms (see left of Fig. 2). For instance, the OD algorithms `ABOD` and `LOF` share a basic operator $k$NN to identify the $k$-th nearest neighbors (see Fig. 3). By identifying these recurring operators for FEDOD, we gain two benefits: (1) since each neural approximator is compatible with FL, combining them can yield FL-enabled OD algorithms, and (2) it reduces design and optimization effort—each basic operator's neural approximator only needs to be designed once.

***Operator categories.*** After reviewing a diverse set of OD algorithms, we categorize basic OD operators into two groups: *convex operators* (§3.1) and *non-convex operators* (§3.2), depending on whether the underlying operator has a convex optimization objective. For an operator with a convex objective, its neural approximation (§4) uses a gradient-based optimization to achieve better convergence properties (e.g., fewer training epochs). In contrast, non-convex OD operators require more specialized design and pose additional challenges for convergence. Additionally, certain *simple operators* do not require any approximation (§3.3). Figure 4 illustrates the OD algorithms supported in the current implementation of FEDOD and the corresponding basic operators by category. Note that operators are independent, and FEDOD can be easily extended to support new OD operators.

## 3.1 CONVEX OD OPERATORS

An operator *convex* if, for every pair of points within its domain, the operator applied to the weighted average of the points is less than or equal to the weighted average of the operator applied to the points Boyd & Vandenberghe (2004). Convex OD operators offer many desirable properties. An important one is that any local minimum of a convex function is also a global minimum, which greatly simplifies the search for a global optimal. Thus, using a gradient-based optimization in a neural network can generally approximate convex operators well via gradient-based optimizations Snyman et al. (2005) and takes less training time. Fig. 4 and Appx. B.1 left discuss a few commonly used convex operators by classical OD algorithms with additional ones including support vector machine (SVM) Tibshirani (1996) and Lasso Cortes & Vapnik (1995).

## 3.2 NON-CONVEX OD OPERATORS

A non-convex operator is one that does not satisfy the convex condition. Handling non-convex operators in optimization is challenging because they often have multiple local minima, and standard gradient-based optimization methods may get stuck in these local minima rather than finding the global minimum, leading to more complexity of neural approximation with gradient-based optimization Boyd & Vandenberghe (2004). Fig. 4 (middle) and Appx. B.2 shows some key operators supported in FEDOD in addition to cosine similarity calculation, and density estimation.

## 3.3 SIMPLE OD OPERATORS

Not all operators necessitate a (neural) approximation. Certain operators do not involve computation and/or optimization, and can thus be directly executed federated. This category includes most sorting and ranking operators. For instance, the $k$NN operator comprises two primary steps: (1) computing

the pairwise distances among samples, and (2) identifying the top $k$ distances per sample. The first step involves computation and should be approximated by a neural network, while the second step, which is about finding the top $k$ values, can be directly applied to the output of the first step, without the need for any approximation. Figure 4 (right) summarizes the simple operators in FEDOD.

# 4 OD APPROXIMATION

## 4.1 NEURAL NETWORK CONVERTER

As existing FL frameworks are designed for neural networks Li et al. (2020a), we propose converting basic OD operators (see §3) each into a neural network. This conversion is theoretically feasible since a neural network is a "universal approximator" Hornik et al. (1989). The choice of neural network approximations, such as architectures, can be flexible, provided that the model capacity is sufficient — the neural networks are adequately complex for approximation (see §5.4.1). Under this criterion, fully-connected multi-layer perceptrons (MLP) Rumelhart et al. (1986) serve as the default choice in FEDOD. We have also explored more recent architectures like Transformers Vaswani et al. (2017) (see §5.4.2). FEDOD's *neural network converter* (NNC) translates basic OD operators in Fig. 4 into neural networks, as depicted in the middle of Fig. 2. Notably, this is the first proposal for neural approximations of general classical OD algorithms such as $k$NN and clustering. With the OD model decomposition and the NNC, a variety of classical OD algorithms can be decomposed into and approximated by neural networks, which can then be trained in existing FL systems.

## 4.2 TRAINING METHODOLOGY

***Local ground truth.*** Even after OD operators are converted to neural networks, training them on each local agent without data sharing requires special considerations. As depicted on the right of Fig. 2, FEDOD's goal is to train a central model $f(\cdot)$, i.e., the neural network generated from NNC as an approximation of the underlying OD operator.

To update $f(\cdot)$, we can average local gradients using the data on each agent, i.e., $\nabla f(\mathbf{X}_1)$, $\nabla f(\mathbf{X}_2)$, a technique known as Federated Averaging (FedAvg)[3] in Eq. (2) McMahan et al. (2017). However, local gradients are not directly computable due to global data dependencies (i.e., the global ground truth of $k$-th nearest neighbors is unavailable).

In FEDOD, we introduce a set of novel *local-update strategies* for each neural OD operator. The key is to design a loss function that allows each neural OD operator to update the model with respect to the *local* data only, to approximate the *global* ground truth.

***Clustering example.*** The goal of $k$-means clustering is to partition $n$ samples across all agents into clusters defined by sample similarity. As described in §3.2, the sample similarity can be measured by pairwise Euclidean distances —samples in the same cluster should be close. For an $n \times d$ input matrix, all $n$ samples are needed for the Euclidean distance calculation (i.e., `cdist`). However, in the FL setting, the $i$-th agent can only access private data $\mathbf{X}_i \in \mathbb{R}^{n_k \times d}$ and the local distance of $\mathbf{D}_i = \mathrm{cdist}(\mathbf{X}_i, \mathbf{X}_i)$. We cannot access the global sample similarity across all agents by only looking at a partition of data on a local agent.

Suppose we use a neural network to approximate $k$-Means without considering privacy preservation, the model may minimize the discrepancy between the predicted cluster assignment and the ground truth cluster assignment (unavailable in our settings). Differently, FEDOD performs local updates without accessing the global ground truth of cluster assignments, as shown in Algorithm 1. The key is to design a loss function *solely based on the local data*.

First, we initialize the central model $f(\cdot)$ with parameters $\mathbf{w}$ and pre-compute the local pairwise distances for all $K$ agents, i.e., $\mathbf{D}_k = \mathrm{cdist}(\mathbf{X}_k, \mathbf{X}_k)$ (lines 1-4). Second, for the $k$-th agent, let $\widehat{\mathbf{c}} = f_k(\mathbf{X}_k)$ denote the predicted cluster labels for the local data by the current model. Given there are $n_k$ samples on the $k$-th agent, our loss defined in Eq. 1 aims to minimize the intra-cluster distance and maximize the inter-cluster distance for each local sample given the predicted cluster labels by $f(\cdot)$ (lines 5-10). Note this only uses the data on the $k$-th agent.

---

[3]Other FL frameworks can also be applied; we use FedAvg as an example.

---

**Algorithm 1** Neural approximation of $k$-Means clustering

---

**Input:** Central neural model $f(\cdot)$ with weights $\mathbf{w}$ and objective function $\mathcal{L}$, $K$ local agents where the $k$-th agent with private data $\mathbf{X}_k \in \mathbb{R}^{n_k \times d}$ in the same $d$-dimension feature space; local neural network models $\{f_1, \ldots, f_K\}$; training budget $T$
**Output:** The trained neural model $f(\cdot)$ across local agents

---

1: **Initialize** the central model $f(\cdot)$
2: **for** each local agent $k = 1$ to $K$ **do**
3:     Compute pairwise local distance matrix $\mathbf{D}_k = \text{cdist}(\mathbf{X}_k, \mathbf{X}_k)$
4: **end for**

---

5: **for** $t = 1$ to $T$ **do**
6:     **for** each local agent $k = 1$ to $K$ **do**
7:         Get current central model weights $\mathbf{w}^t$ and initialize the local model $f_k$ to it
8:         Predict the cluster labels on the local data $\widehat{\mathbf{c}} = f_k(\mathbf{X}_k)$
9:         Compute *local* loss $\mathcal{L}_k = (\widehat{\mathbf{c}}, \mathbf{D}_k)$ and gradient $\nabla \mathbf{w}_k^{(t)}$ to minimize the intra-cluster and maximize the inter-cluster distance given the predicted cluster labels by Eq. (1)
10:     **end for**
11:     Update the central model $\mathbf{w}^{(t+1)}$ by FedAvg in Eq. (2)
12: **end for**

---

$$\mathcal{L}(\widehat{\mathbf{c}}, \mathbf{D}_k)) = \sum_{i=1}^{n_k} \sum_{j \neq i} (-\underbrace{\sum_{\widehat{\mathbf{c}}_i = \widehat{\mathbf{c}}_j} \mathbf{D}_{i,j}}_{\text{min. intra dist.}} + \underbrace{\sum_{\widehat{\mathbf{c}}_i \neq \widehat{\mathbf{c}}_j} \mathbf{D}_{i,j}}_{\text{max. inter dist.}}) \tag{1}$$

At $t$-th iteration, FEDOD iterates over all the agents to aggregate the local gradients $\mathbf{w}_1^{(t)}, \ldots, \mathbf{w}_K^{(t)}$ (line 11).; one straightforward way is to use FedAvg as shown in Eq. (2).

$$\mathbf{w}^{(t+1)} := \mathbf{w}^{(t)} + \frac{1}{n} \sum_{k=1}^{K} n_k \cdot \nabla \mathbf{w}_k^{(t)} \tag{2}$$

In this way, we could update the central model without accessing any global samples as well as ground truth labels. We design a local loss function For each supported operator in FEDOD.

## 4.3 ADVANTAGES OF FEDOD

***Anytime inference.*** An anytime algorithm can return a valid solution even if it is not fully complete Cutkosky (2019), which is particularly useful in real-time OD applications. These algorithms typically present a trade-off between computational resources, like time or memory, and the quality of the solution; for example, longer training time results in better accuracy. Neural networks are naturally anytime algorithms due to their iterative training nature—we can make predictions at any time during the training process. In contrast, most classical OD algorithms (see Table A1) cannot make predictions until the training is fully complete, limiting their applicability in time-critical applications. However, FEDOD's neural approximation techniques convert classical OD algorithms to anytime algorithms, thus enhancing their flexibility.

***Fast inference and optimization techniques.*** Existing OD algorithms often have high inference cost due to distance calculation and/or density estimation. By employing neural approximation in FEDOD, the inference time can be largely reduced to a single forward pass with the data. Thus, for large datasets with high feature dimensions, FEDOD offers shorter inference time. Additionally, any neural network optimization techniques, such as parallelization Schneider et al. (2021); Wang et al. (2023), quantization Zhao et al. (2023), also apply to shallow OD algorithms via FEDOD. See §5.3 for a detailed comparison of inference times.

***Generality and extensibility.*** Thanks to the generality of basic operators in FEDOD, it can be easily extended to support ML algorithms beyond OD, such like classification. Moreover, integrating new operators into FEDOD is straightforward, requiring just two simple steps. First, we need to

understand the general properties of the operator to be added, including its convexity, inputs, and outputs. Second, we need to design a local loss function that can be computed solely based on the local data, as described in §4.2. Our experience of developing FEDOD shows that it generally takes a few hours to add a new OD operator and its neural approximation.

## 5 EXPERIMENTS

Our experiments aim to answer the following questions: ($i$) How does the detection performance of FEDOD compare to baselines, both with and without privacy preservation? (§5.2) ($ii$) How efficient and scalable is FEDOD when handling larger datasets? (§5.3) ($iii$) How do the designs in FEDOD impact its effectiveness? (§5.4)

### 5.1 EXPERIMENT SETUP

***Datasets.*** Appx Table C2 displays over 21 real-world OD datasets used in this study, primarily sourced from two popular repositories, i.e., DAMI Campos et al. (2016) and ODDS Rayana (2016). These datasets have been widely employed in OD research Tran et al. (2020); Schmidl et al. (2022); Han et al. (2022); Zhao et al. (2023).

***OD algorithms and operators.*** We include five diverse OD algorithms to show the effectiveness of FEDOD: (1) Distance-based $k$NN for OD Ramaswamy et al. (2000); (2) Density-based local outlier factor (LOF) Breunig et al. (2000); (3) Linear PCA for OD Shyu et al. (2003); (4) Clustering-based OD method (CBLOF) He et al. (2003); and (5) Tree-based isolation forest (iForest) Liu et al. (2008).

***Implementation and environment.*** FEDOD is implemented on top of PyTorch Paszke et al. (2019), where most algorithms only depend on 2-3 operators with low complexity. All the experiments are performed on an Amazon EC2 p3.2xlarge cluster with an Intel Xeon CPU, 61GB DRAM, and an NVIDIA Tesla V100 GPU with 16GB RAM.

***Baselines.*** As the first work for privacy-preserving OD with shallow methods, FEDOD is compared with (1) *ground truth* where the access to all samples is assumed (no privacy preservation at all); we use PyOD Zhao et al. (2019) to get the results and (2) *direct* results where we train $K$ individual models (one per local dataset) and concatenate the results (data privacy is preserved).

***Evaluation metrics.*** Enabled by model approximation, FEDOD's goal is to achieve similar performance as the *ground truth* baseline while preserving privacy. With that in mind, we measure the performance difference[4] to the ground truth—the smaller the difference, the better the method.

### 5.2 END-TO-END COMPARISONS

Appx. Table C3 illustrates that FEDOD achieves performance close to the ground truth and surpasses the *direct* baseline across all five OD algorithms with differing characteristics. Specifically, FEDOD displays less than 5% ROC-AUC difference from the ground truth across all five OD algorithms (1.79%, 3.93%, 2.05%, 1.96%, and 4.70% for $k$NN, LOF, PCA, CBLOF, and iForest, respectively), whereas the *direct* method shows up to a 19% performance difference. It is worth noting that FEDOD is 11 times better than direct on $k$NN on average (1.79% vs. 18.92%).

Convex operators seem to exhibit smaller approximation differences. As discussed in §3.1, PCA (Table C3c) presents relatively minor performance differences (2.05%) and variations (2.44%). This can be attributed to the favorable convergence properties of convex operators, whereas non-convex operators like LOF (Table C3b, 3.93% $\pm$ 4.17%) and iForest (Table C3e, 4.70% $\pm$ 4.72%) may present much larger differences and variations (which are less desirable).

### 5.3 SCALABLITY OF FEDOD

We evaluate the scalability of FEDOD across datasets of varying sizes (ranging from 5,000 to 320,000 samples with 1,000 features), thereby simulating scenarios of OD on high-dimensional, large datasets. Fig. 5 illustrates the inference time of FEDOD (in black) and *direct* baseline (in red).

---

[4]We use the area under the Receiver Operating Characteristic curve (ROC) as the performance; this can be substituted with any other measure of interest.

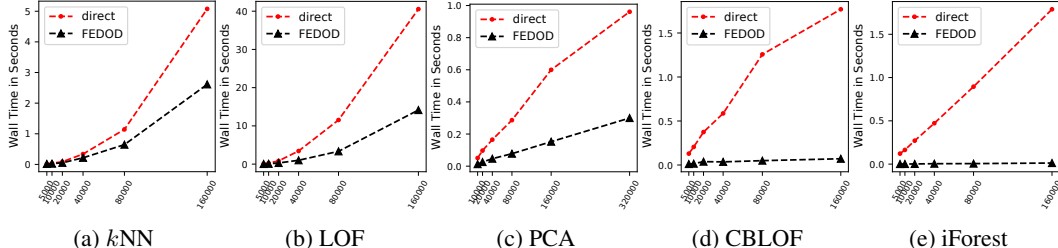

|      |      |      |      |      |
| ---- | ---- | ---- | ---- | ---- |
| (a) $k$NN | (b) LOF | (c) PCA | (d) CBLOF | (e) iForest |

Figure 5: Scalability plot of algorithms in FEDOD, where it scales well with increasing samples. FEDOD exhibits superior efficiency compared to *direct* across all datasets, particularly larger ones. For example, FEDOD achieves a $10\times$ speed-up over the direct method on the `CBLOF` and `iForest` datasets. One of the primary reasons for this improvement is that the *direct* method needs to invoke multiple models for inference, while FEDOD relies solely on a single central model. Moreover, FEDOD only requires a single pass of the neural network, which is significantly more economical than the *direct* method, which relies on distance and/or density estimation.

FEDOD demonstrates robust scalability with larger datasets. As shown in Fig. 5, the inference time scales linearly about the number of inference samples. This is attributed to the fact that the network parameters are fixed—the inference time is determined solely by the number of inference samples.

### 5.4 ABLATION STUDIES AND ADDITIONAL ANALYSIS

#### 5.4.1 EFFECT OF MODEL CAPACITY

We assess model capacity by modifying the number of hidden neurons (x-axis) and the number of layers (y-axis) in the neural approximation of $k$NN, and compare their performance variance with the *ground truth* in Fig. 6. The results indicate that FEDOD's performance is fairly insensitive to the neural model capacity, provided it is adequately large, with all variations remaining under 3%. Thus, we employ 64 hidden neurons and 2 layers across all experiments. Utilizing a comparatively small model curbs both training and inference costs. Also note that the basic operators are generally simple, and thus small neural network could be more cost-effective.

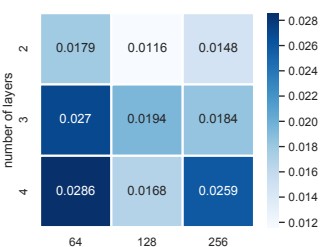

Figure 6: Ablation studies on model capacity (i.e., size of neural networks) of approximation: (x-axis) the number of hidden neurons and (y-axis) the number of layers. We show the avg. performance diff. (smaller the better) across all datasets on $k$NN (Table C3a); FEDOD is insensitive model capacity with small differences with varying sizes of neural networks.

#### 5.4.2 THE CHOICE OF BACKBONE IN FEDOD

We evaluate the performance of using a simple MLP versus the more recent transformers Vaswani et al. (2017) as the backbone of the neural approximator in §4. Transformers, capable of capturing attention between both data points and features, have demonstrated strong performance in recent ADBench studies Han et al. (2022). This might help us establish a relationship that correlates input directly to output, based not only on feature combinations but also on other data points. Here we utilize a transformer model with two self-attention heads and a stacking depth of two.

Appx. Table C3f presents the performance of $k$NN using both MLP (Table C3a) and transformers (Table C3f). We observe that MLP outperforms transformers in 14 out of 21 datasets, while the average performances are comparable (1.79% vs. 2.77%). Note that both are superior to the *direct* baselines (18.92%). One reason for the lower performance of transformers might be their complexity and sensitivity to hyperparameters Chen et al. (2020). This affirms the choice of MLP as the default backbone for FEDOD, given its simplicity, speed, and robustness.

## 6 LIMITATIONS AND FUTURE DIRECTIONS

***Automated network conversion.*** At present, the conversion of operators in FEDOD is carried out manually, a process that could be further automated using meta-learning Vanschoren (2018).

For instance, we could train multiple neural approximators on existing datasets to evaluate their performance. When presented with a new dataset, we identify the most similar historical dataset and transfer the optimal neural configurations Zhao et al. (2021) in a zero-shot manner. Also, network parameters from similar historical datasets could be transferred Scott et al. (2018), thus reducing the training cost.

***Inference optimization.*** On top of the already efficient inference, FEDOD may further leverage acceleration techniques for neural networks, including quantization Hubara et al. (2017), distributed and multi-GPU learning Jia et al. (2017), etc. However, it is worth noting that further acceleration may be at the cost of larger performance differences.

## 7 RELATED WORK

***Federated learning for OD.*** OD is a crucial task in a variety of applications such as credit card fraud detection Zhong et al. (2020), cybersecurity Mothukuri et al. (2021), healthcare Gupta et al. (2021), and environmental anomaly detection Chandola et al. (2009). In many circumstances, due to privacy constraints or regulatory requirements, data cannot be shared, rendering traditional OD methods untenable. There exists a huge need for privacy-preserving OD methods that can operate without sharing data, thereby preventing potential misuse of sensitive info Sater & Hamza (2021).

Federated learning (FL), as a leading privacy-preserving framework, has been applied to various *deep* OD algorithms. These algorithms employ different neural architectures, including recurrent neural networks (RNN) Nguyen et al. (2019); Mothukuri et al. (2021), autoencoders Pei et al. (2022), LSTM Sater & Hamza (2021), and CNN Astillo et al. (2022). There are OD applications in diverse domains such as healthcare Gupta et al. (2021); Astillo et al. (2022), internet of things (IoT) Nguyen et al. (2019); Mothukuri et al. (2021), network security Pei et al. (2022), and smart buildings Sater & Hamza (2021). Most of these works primarily focus on horizontal FL, where different agents (e.g., mobile devices or workstations) hold a partition of samples from the same feature space. FL for deep OD typically follows the following procedures Sater & Hamza (2021). Initially, a global neural network OD model is initialized. Then, local datasets are utilized to update the model parameters through local training using techniques such as stochastic gradient descent (SGD). Finally, the updated model parameters from each device are aggregated to update the global model for OD, such as Federated Averaging (FedAvg) McMahan et al. (2017).

However, most classical OD algorithms cannot leverage existing FL frameworks due to inter-sample data dependency as we elaborate in the introduction. Table A1 summarizes a diverse group of classical OD algorithms, many of which lack a straightforward FL solution due to data dependency. Hence, FEDOD is designed to address this gap while accelerating inference.

***Neural approximation for OD.*** Neural networks are widely recognized for their remarkable function approximation capabilities Goodfellow et al. (2016). This is predominantly due to their ability to model complex, high-dimensional, and nonlinear relationships inherent in data. The Universal Approximation Theorem provides theoretical support to this notion by asserting that a feedforward network with a single hidden layer containing a finite number of neurons can approximate continuous functions under specific conditions Hornik et al. (1989). Also, deep neural networks can represent complex functions more compactly. This allows them to model complex patterns in data, making them useful in various applications such as language translation and autonomous driving LeCun et al. (2015). In FEDOD, we employ neural networks to approximate OD operators for the dual purposes of *preserving privacy* and *accelerating inference*.

## 8 CONCLUSION

We introduce FEDOD, a novel system designed to overcome challenges associated with privacy preservation and efficiency in outlier detection applications. FEDOD enables popular federated learning paradigm and extends its benefits to over 20 shallow (non-neural-network) OD algorithms. The key steps include decomposing OD algorithms into shared operators for neural approximation, after which federated learning becomes compatible with the operators. Through extensive experiments on more than 30 datasets and 5 diverse detection algorithms, we demonstrate that FEDOD can efficiently approximate OD algorithms while ensuring privacy. Future work can extend FEDOD's support to more other ML tasks and optimize it by neural network acceleration techniques.

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
