Table A1: Key classical OD algorithms and whether they are inter-sample dependent (i.e., requiring other samples for computation) and inter-feature dependent (i.e., requiring other features for computation). These methods are generally costly in inference, where $n$ and $n_t$ denote the numbers of training and inference samples, and $d$ is the number of features. Ensemble algorithms' inference time depends on base estimators' complexity (the last three rows).

| Algorithm | Inter-sample | Inter-feature | Infer time |
|---|:---:|:---:|:---:|
| $k$NN (OD) Ramaswamy et al. (2000) | ✓ | ✓ | $n \cdot n_t \cdot d$ |
| COF Tang et al. (2002) | ✓ | ✓ | $n \cdot n_t \cdot d$ |
| LOF Breunig et al. (2000) | ✓ | ✓ | $n \cdot n_t \cdot d$ |
| LOCI Papadimitriou et al. (2003) | ✓ | ✓ | $n \cdot n_t \cdot d$ |
| SOD Kriegel et al. (2009) | ✓ | ✓ | $n \cdot n_t \cdot d$ |
| CBLOF He et al. (2003) | ✓ | ✓ | $n \cdot n_t \cdot d$ |
| HBOS Goldstein & Dengel (2012) | ✓ | ✗ | $n_t \cdot d$ |
| COPOD Li et al. (2020b) | ✓ | ✗ | $n_t \cdot d$ |
| ECOD Li et al. (2022) | ✓ | ✗ | $n_t \cdot d$ |
| PCA Shyu et al. (2003) | ✓ | ✓ | $n_t \cdot d$ |
| OCSVM Schölkopf et al. (2001) | ✓ | ✓ | $n_t \cdot d$ |
| LODA Pevný (2016) | ✗ | ✗ | N/A |
| FB Lazarevic & Kumar (2005) | ✓ | ✗ | N/A |
| iForest Liu et al. (2008) | ✗ | ✗ | N/A |

## B    DETAILS ON MODEL DECOMPOSITION

### B.1    CONVEX OPERATORS

***Linear regression*** is a popular statistical method that investigates the linear relationship between predictor variables James et al. (2013), which can be optimized by gradient descent via minimizing a convex mean squared loss. For OD, it can identify outlying samples by examining the residuals Aggarwal (2013), suggesting that these data points do not follow the same pattern as most of the data.

***Singular value decomposition (SVD)*** is a matrix decomposition technique used extensively in ML Golub & Van Loan (2013). Notably, SVD can be transformed into convex problems for solving Boyd & Vandenberghe (2004). Under the context of OD, SVD is an important part of *principal component analysis* (PCA), which identifies outliers by projecting the data onto the subspace spanned by the leading singular vectors; points far away from this subspace's origin can be considered outliers Aggarwal (2013).

### B.2    NON-CONVEX OPERATORS

***kNN*** is a non-parametric, instance-based ML method Fix & Hodges (1989). The central concept is that similar data samples should yield similar outputs. Sample similarity is often gauged by distances, which does not constitute a convex problem since it is not an optimization task.

***Clustering*** endeavors to partition samples into a predefined number of clusters based on certain criteria, e.g., proximity to the cluster center. For instance, the popular $k$-Means method MacQueen et al. (1967) is not a convex problem, as its objective is the sum of squared distances from each point to the centroid of its assigned cluster.

***Decision tree*** is a tree-like model of decisions, where each internal node corresponds to a feature in the data, each branch signifies a decision rule, and each leaf node represents an outcome Quinlan (1986). It is known that the problem of finding an optimal decision tree is NP-complete and non-convex, with a combinatorial explosion of potential tree structures (and hence many local optima) as the number of features increases Laurent & Rivest (1976).

## C EXPERIMENT DETAILS

### C.1 DATASET AND CODE

Appx Table C2 displays over 21 real-world OD datasets used in this study, primarily sourced from two popular repositories, i.e., DAMI Campos et al. (2016) and ODDS Rayana (2016). These datasets have been widely employed in OD research Tran et al. (2020); Schmidl et al. (2022); Han et al. (2022); Zhao et al. (2023).

Table C2: Twenty-one real-world OD datasets used in the experiments. We also create and use synthetic datasets throughout the experiments to demonstrate the results on larger datasets.

| Dataset | Source | #Samples | #Dims | %Outlier |
|---|---|---|---|---|
| Annthyroid | DAMI | 7129 | 21 | 7.49 |
| Cardio | DAMI | 2114 | 21 | 22.04 |
| Glass | DAMI | 214 | 7 | 4.21 |
| Heart | DAMI | 270 | 13 | 44.44 |
| InternetAds | DAMI | 1966 | 1555 | 18.72 |
| PageBlocks | DAMI | 5393 | 10 | 9.46 |
| Pima | DAMI | 768 | 7 | 34.9 |
| SpamBase | DAMI | 4207 | 57 | 39.91 |
| Stamps | DAMI | 340 | 9 | 9.12 |
| WBC | DAMI | 223 | 9 | 4.48 |
| ionosphere | ODDS | 351 | 33 | 35.9 |
| mammog. | ODDS | 11183 | 6 | 2.32 |
| mnist | ODDS | 7603 | 100 | 9.21 |
| pima | ODDS | 768 | 8 | 34.9 |
| satellite | ODDS | 6435 | 36 | 31.64 |
| satimage-2 | ODDS | 5803 | 36 | 1.22 |
| speech | ODDS | 3686 | 400 | 1.65 |
| thyroid | ODDS | 3772 | 6 | 2.47 |
| vowels | ODDS | 1456 | 12 | 3.43 |
| wbc | ODDS | 378 | 30 | 5.56 |

The demonstration code is available at anonymous Google Drive: `https://tinyurl.com/fedod2023`.

### C.2 END-TO-END RESULTS

Appx. Table C3 illustrates that FEDOD achieves performance close to the ground truth and surpasses the *direct* baseline across all five OD algorithms with differing characteristics. Specifically, FEDOD displays less than 5% ROC-AUC difference from the ground truth across all five OD algorithms (1.79%, 3.93%, 2.05%, 1.96%, and 4.70% for $k$NN, LOF, PCA, CBLOF, and iForest, respectively), whereas the *direct* method shows up to a 19% performance difference. It is worth noting that FEDOD is 11 times better than direct on $k$NN on average (1.79% vs. 18.92%).

Table C3: ROC-AUC comparison of *ground truth*, *ours*, and *direct* across five diverse OD algorithms (subtable C3a to C3e). Across all five OD methods, *ours* consistently shows competitive performance with small differences to *ground truth* (up to 5% on average), outperforming the *direct* method in most cases. By maintaining a lower Avg. $|\Delta|$ values to the ground truth (up to 11× better than *direct*), FEDOD demonstrates its efficacy and robustness across OD scenarios. Subtable C3f shows the ablation study on using MLP (Ours) vs. transformers (Transf.) as the backbone of FEDOD; see more detailed analysis in §5.4.2.

| Dataset | Ground truth | Ours | ΔOurs | Direct | ΔDirect |
|---|---|---|---|---|---|
| Annthyroid | 0.6616 | 0.6188 | -6.47% | 0.5389 | -18.54% |
| Cardio | 0.4833 | 0.4923 | 1.87% | 0.6503 | 34.56% |
| Glass | 0.7619 | 0.7381 | -3.12% | 0.4762 | -37.50% |
| Heart | 0.5628 | 0.5411 | -3.85% | 0.4776 | -15.13% |
| InternetAds | 0.6868 | 0.6770 | -1.44% | 0.6623 | -3.57% |
| PageBlocks | 0.9128 | 0.9164 | 0.39% | 0.8083 | -11.45% |
| Pima | 0.7734 | 0.7831 | 1.26% | 0.6817 | -11.85% |
| SpamBase | 0.5897 | 0.5938 | 0.70% | 0.5358 | -9.13% |
| Stamps | 0.8857 | 0.9111 | 2.87% | 0.4508 | -49.10% |
| WBC | 0.9773 | 0.9773 | 0.00% | 1.0000 | 2.33% |
| arrhythmia | 0.8017 | 0.8006 | -0.13% | 0.8439 | 5.26% |
| ionosphere | 0.9294 | 0.9145 | -1.61% | 0.6728 | -27.61% |
| mammog. | 0.8505 | 0.8807 | 3.56% | 0.6484 | -23.77% |
| mnist | 0.8462 | 0.8544 | 0.97% | 0.7249 | -14.33% |
| pima | 0.7372 | 0.7497 | 1.69% | 0.6534 | -11.36% |
| satellite | 0.6879 | 0.7167 | 4.19% | 0.4711 | -31.52% |
| satimage-2 | 0.9826 | 0.9966 | 1.42% | 0.9420 | -4.14% |
| speech | 0.5750 | 0.5678 | -1.25% | 0.6266 | 8.99% |
| thyroid | 0.9720 | 0.9677 | -0.43% | 0.8454 | -13.02% |
| vowels | 0.9731 | 0.9735 | 0.03% | 0.4555 | -53.19% |
| wbc | 0.9310 | 0.9333 | 0.26% | 0.8286 | -11.00% |
| Avg. $\|\Delta\|$ | N/A | 1.79%± 1.67% | | 18.92%± 14.73% | |

(a) $k$NN (distance-based)

| Dataset | Ground truth | Ours | ΔOurs | Direct | ΔDirect |
|---|---|---|---|---|---|
| Annthyroid | 0.6737 | 0.6300 | -6.49% | 0.6134 | -8.94% |
| Cardio | 0.5146 | 0.5121 | -0.48% | 0.5733 | 11.41% |
| Glass | 0.8538 | 0.8641 | 1.20% | 0.8638 | 1.17% |
| Heart | 0.5467 | 0.5651 | 3.35% | 0.5267 | -3.66% |
| InternetAds | 0.6005 | 0.6018 | 0.22% | 0.6221 | 3.60% |
| PageBlocks | 0.6825 | 0.7192 | 5.38% | 0.9313 | 36.45% |
| Pima | 0.6311 | 0.6487 | 2.79% | 0.6556 | 3.89% |
| SpamBase | 0.4610 | 0.4777 | 3.62% | 0.4982 | 8.06% |
| Stamps | 0.6227 | 0.6863 | 10.22% | 0.6409 | 2.93% |
| WBC | 0.7873 | 0.7593 | -3.55% | 0.7910 | 0.47% |
| arrhythmia | 0.7264 | 0.7416 | 2.09% | 0.7420 | 2.15% |
| ionosphere | 0.9065 | 0.9083 | 0.20% | 0.8976 | -0.98% |
| mammog. | 0.7095 | 0.7003 | -1.30% | 0.8079 | 13.86% |
| mnist | 0.6451 | 0.6591 | 2.17% | 0.8565 | 32.77% |
| pima | 0.5945 | 0.5946 | 0.02% | 0.6125 | 3.02% |
| satellite | 0.5289 | 0.5524 | 4.43% | 0.6114 | 15.60% |
| satimage-2 | 0.6186 | 0.5235 | -15.37% | 0.9899 | 60.02% |
| speech | 0.5081 | 0.4753 | -6.45% | 0.4545 | -10.55% |
| thyroid | 0.6076 | 0.6800 | 11.91% | 0.9729 | 60.11% |
| vowels | 0.9394 | 0.9515 | 1.29% | 0.9354 | -0.42% |
| wbc | 0.9553 | 0.9561 | 0.08% | 0.9526 | -0.29% |
| Avg. $\|\Delta\|$ | N/A | 3.93%± 4.17% | | 13.35%± 18.39% | |

(b) LOF (density-based)

| Dataset | Ground truth | Ours | ΔOurs | Direct | ΔDirect |
|---|---|---|---|---|---|
| Annthyroid | 0.5622 | 0.5864 | 4.30% | 0.5718 | 1.71% |
| Cardio | 0.7440 | 0.7496 | 0.76% | 0.7270 | -2.28% |
| Glass | 0.6195 | 0.6450 | 4.11% | 0.6515 | 5.16% |
| Heart | 0.5881 | 0.5727 | -2.62% | 0.6253 | 6.33% |
| InternetAds | 0.6146 | 0.6144 | -0.04% | 0.6017 | -2.11% |
| PageBlocks | 0.9047 | 0.8982 | -0.71% | 0.7949 | -12.13% |
| Pima | 0.6698 | 0.6339 | -5.35% | 0.6364 | -4.98% |
| SpamBase | 0.5506 | 0.5444 | -1.12% | 0.4572 | -16.95% |
| Stamps | 0.8989 | 0.8976 | -0.15% | 0.9217 | 2.53% |
| WBC | 0.9826 | 0.9803 | -0.24% | 0.9836 | 0.10% |
| arrhythmia | 0.7749 | 0.7806 | 0.73% | 0.7699 | -0.65% |
| ionosphere | 0.7963 | 0.8339 | -0.03% | 0.7822 | -1.78% |
| mammog. | 0.8835 | 0.8711 | -1.40% | 0.8740 | -1.08% |
| mnist | 0.8501 | 0.8501 | 0.00% | 0.7336 | -13.70% |
| pima | 0.6585 | 0.6113 | -7.17% | 0.6113 | -7.17% |
| satellite | 0.6019 | 0.6312 | 4.87% | 0.6207 | 3.13% |
| satimage-2 | 0.9771 | 0.9776 | 0.05% | 0.9078 | -7.10% |
| speech | 0.4692 | 0.4692 | 0.00% | 0.5729 | 22.10% |
| thyroid | 0.9455 | 0.9305 | -1.59% | 0.9452 | -0.04% |
| vowels | 0.6103 | 0.6072 | -0.50% | 0.2494 | -59.14% |
| wbc | 0.9352 | 0.8665 | -7.35% | 0.9406 | 0.58% |
| Avg. $\|\Delta\|$ | N/A | 2.05% ± 2.44% | | 8.13%± 13.13% | |

(c) PCA (linear)

| Dataset | Ground truth | Ours | ΔOurs | Direct | ΔDirect |
|---|---|---|---|---|---|
| Annthyroid | 0.5815 | 0.6020 | 3.53% | 0.5569 | -4.23% |
| Cardio | 0.5653 | 0.5764 | 1.97% | 0.5839 | 3.29% |
| Glass | 0.8885 | 0.8615 | -3.03% | 0.9064 | 2.02% |
| Heart | 0.6003 | 0.5826 | -2.95% | 0.6220 | 3.62% |
| InternetAds | 0.6297 | 0.6292 | -0.09% | 0.6329 | 0.51% |
| PageBlocks | 0.8799 | 0.8875 | 0.86% | 0.8834 | 0.39% |
| Pima | 0.6376 | 0.6276 | -1.56% | 0.6344 | -0.50% |
| SpamBase | 0.5496 | 0.5535 | 0.70% | 0.5493 | -0.06% |
| Stamps | 0.6419 | 0.6879 | 7.17% | 0.7687 | 19.76% |
| WBC | 0.9562 | 0.9674 | 1.17% | 0.9655 | 0.98% |
| arrhythmia | 0.7334 | 0.7668 | 4.55% | 0.7358 | 0.32% |
| ionosphere | 0.9237 | 0.8553 | 1.12% | 0.9110 | -1.38% |
| mammog. | 0.7922 | 0.7774 | -1.86% | 0.7966 | 0.56% |
| mnist | 0.8355 | 0.8460 | 1.26% | 0.8345 | -0.12% |
| pima | 0.6245 | 0.6222 | -0.37% | 0.6200 | -0.71% |
| satellite | 0.7354 | 0.7186 | -2.28% | 0.7313 | -0.55% |
| satimage-2 | 0.9980 | 0.9921 | -0.59% | 0.9905 | -0.75% |
| speech | 0.4666 | 0.4652 | -0.30% | 0.4681 | 0.33% |
| thyroid | 0.9262 | 0.9542 | 3.02% | 0.9283 | 0.22% |
| vowels | 0.8771 | 0.8558 | -2.43% | 0.9059 | 3.28% |
| wbc | 0.9383 | 0.9352 | -0.34% | 0.9826 | 4.72% |
| Avg. $\|\Delta\|$ | N/A | 1.96% ± 1.70% | | 2.30%± 2.46% | |

(d) CBLOF (clustering-based)

| Dataset | Ground truth | Ours | ΔOurs | Direct | ΔDirect |
|---|---|---|---|---|---|
| Annthyroid | 0.6178 | 0.6620 | 7.15% | 0.5961 | -3.51% |
| Cardio | 0.6794 | 0.6987 | 2.83% | 0.8647 | 27.27% |
| Glass | 0.7381 | 0.6667 | -9.68% | 0.8435 | 14.29% |
| Heart | 0.5281 | 0.5988 | 13.39% | 0.7394 | 40.00% |
| InternetAds | 0.7076 | 0.7468 | 5.55% | 0.7960 | 12.50% |
| PageBlocks | 0.9041 | 0.9017 | -0.26% | 0.9589 | 6.06% |
| Pima | 0.7398 | 0.7388 | -0.12% | 0.8190 | 10.71% |
| SpamBase | 0.6419 | 0.5934 | -7.55% | 0.7530 | 17.31% |
| Stamps | 0.9016 | 0.9175 | 1.76% | 1.1592 | 28.57% |
| WBC | 1.0000 | 1.0000 | 0.00% | 1.0309 | 3.09% |
| arrhythmia | 0.8439 | 0.8112 | -3.87% | 1.1685 | 38.46% |
| ionosphere | 0.8264 | 0.7965 | -3.62% | 0.7128 | -13.75% |
| mammog. | 0.8636 | 0.8588 | -0.56% | 0.8750 | 1.32% |
| mnist | 0.7809 | 0.8065 | 3.28% | 0.8752 | 12.07% |
| pima | 0.7185 | 0.7423 | 3.32% | 0.8230 | 14.55% |
| satellite | 0.6996 | 0.6824 | -2.46% | 0.9740 | 39.22% |
| satimage-2 | 0.9947 | 0.9960 | 0.13% | 1.5789 | 58.73% |
| speech | 0.5984 | 0.5385 | -10.01% | 0.6250 | 4.44% |
| thyroid | 0.9906 | 0.9788 | -1.20% | 1.1697 | 18.07% |
| vowels | 0.7810 | 0.6438 | -17.56% | 0.9001 | 15.25% |
| wbc | 0.9095 | 0.9500 | 4.45% | 1.0376 | 14.08% |
| Avg. $\|\Delta\|$ | N/A | 4.70%± 4.72% | | 18.73%± 14.87% | |

(e) iForest (tree-based)

| Dataset | Ground truth | Ours | ΔOurs | Transf. | ΔTransf. |
|---|---|---|---|---|---|
| Annthyroid | 0.6616 | 0.6188 | 6.47% | 0.5981 | 9.59% |
| Cardio | 0.4833 | 0.4923 | 1.86% | 0.5098 | 5.48% |
| Glass | 0.7619 | 0.7381 | 3.12% | 0.7857 | 3.13% |
| HeartDisease | 0.5628 | 0.5411 | 3.86% | 0.5426 | 3.59% |
| InternetAds | 0.6868 | 0.6770 | 1.43% | 0.6779 | 1.30% |
| PageBlocks | 0.9128 | 0.9164 | 0.39% | 0.8955 | 1.90% |
| Pima | 0.7734 | 0.7831 | 1.25% | 0.7563 | 2.21% |
| SpamBase | 0.5897 | 0.5938 | 0.70% | 0.5994 | 1.64% |
| Stamps | 0.8857 | 0.9111 | 2.87% | 0.9079 | 2.51% |
| WBC | 0.9773 | 0.9773 | 0.00% | 1.0000 | 2.32% |
| arrhythmia | 0.8017 | 0.8006 | 0.14% | 0.8323 | 3.81% |
| ionosphere | 0.9294 | 0.9145 | 1.60% | 0.9032 | 2.82% |
| mammog. | 0.8505 | 0.8807 | 3.55% | 0.8761 | 3.01% |
| mnist | 0.8462 | 0.8544 | 0.97% | 0.8078 | 4.53% |
| pima | 0.7372 | 0.7497 | 1.70% | 0.7337 | 0.47% |
| satellite | 0.6879 | 0.7167 | 4.19% | 0.6894 | 0.21% |
| satimage-2 | 0.9826 | 0.9966 | 1.42% | 0.9612 | 2.18% |
| speech | 0.5750 | 0.5678 | 1.25% | 0.5719 | 0.53% |
| thyroid | 0.9720 | 0.9677 | 0.44% | 0.9632 | 0.90% |
| vowels | 0.9731 | 0.9735 | 0.04% | 0.9428 | 3.12% |
| wbc | 0.9310 | 0.9333 | 0.25% | 0.9571 | 2.81% |
| Avg. $\|\Delta\|$ | N/A | 1.79%± 1.67% | | 2.77%± 2.00% | |

(f) ablation $k$NN (MLP vs. transformers)