# OpenReview forum: "FedOD: Federated Outlier Detection via Neural Approximation"
_ICLR.cc/2024/Conference — Submitted to ICLR 2024_

### Official Review · Reviewer_4zZc · 2023-10-13

**Soundness:** 1 poor
**Presentation:** 1 poor
**Contribution:** 1 poor
**Rating:** 1
**Confidence:** 4

**Summary:**

The paper is concerned with outlier detection in a federated setting (where data sharing is forbidden). In particular, it focuses on clustering problems. Performance results are provided on a variety of datasets and compared with a baseline ‘direct’ method.

**Strengths:**

The paper addresses important problems: outlier detection across private datasets and clustering across private datasets. Code has been posted to an anonymous website.

**Weaknesses:**

The proposed contribution is minor, the proposed algorithm is not clear, discussion of the state-of-the are is inadequate, and the experimental section lacks key details, and lacks comparison with state-of-the-art algorithms.  The basic difference from a standard federated learning algorithm for classification is in the choice of the local loss function in (1) (which is referred to in the paper as “using specialized local update strategies”). See "Questions" for details.

**Questions:**

I have concerns and questions regarding the algorithm, the experimental setup, and the positioning wrt SoA.

Algorithm:

- It is not clear if the paper proposes to solve a general clustering problem, or a binary one (outlier vs. non-outlier)?

- The “Algorithm” and corresponding discussion in the main paper are about clustering. Do the authors also use NNs to approximate linear regression, SVD, and decision trees? If so, what are the corresponding architectures? If not, how is FedOD used with them?

- The criticism of the “classical algorithms” is caveated with “most” and “many” – So which of the “classical” algorithms are amenable to federated learning? Which of them have “low complexity”?

-The so-called “decomposition into basic operations” is merely to identify them as convex or non-convex. The novel contribution is unclear. And it is not clear how the proposed algorithm solves the local minima issue in the non-convex case.

-Sec 3.3: How does one sort and rank across datasets without sharing datasets?

- Why does the proposed algorithm apply only to shallow OD and not to deep OD algorithms?

- Section 8 states: “extends its benefits to over 20 shallow (non-neural-network) OD algorithms”. What are these 20 algorithms? Why are they labeled “shallow”?

The experimental section is unacceptable:

- The choice of the “direct” methods needs justification. Why is this the best state-of-the-art method for comparison? As explained in the paper, this involves training K separate models (one at each user) and then “concatenating” them; so, there is no federated learning in the “direct” case. Insufficient comparison with state-of-the-art.

- The setting is supposed to be ‘federated’, but the experimental section provides NO details of how many clients were in the setting, and how data was distributed across the clients (iid, non-iid), how many training epochs were used, etc.

- The model architecture is not clear. A fully connected MLP is mentioned at one point, but it is not clear how many layers were in the model, and the dimensions of each layer.

- Inference time comparisons are provided for the proposed method vs. classical methods. Absent is a table showing the training times and training complexity.

- Section C.1 states: “We also create and use synthetic datasets throughout the experiments to demonstrate the results on larger datasets.” – but results of performance on synthetic datasets are not reported. These would be very insightful.

- Some of the datasets described in Table C.2 are a bit odd. The Heart dataset is reported to have 44.44% outliers! Does this, in fact, mean that the Heart dataset is essentially a balanced dataset for a binary classification problem vs. say the satimae-2 dataset that has only 1.2% outliers. Am assuming that the ground truth on the outliers was provided with the datasets – if not, please clarify how the ground truth was established.

- Table C3: What does “AUC” for ground truth mean? And what does it mean to say that the proposed algorithm is better than ground truth (\Delta >0)? The row showing ‘average \Delta’ is meaningless.

- AUC represents a gross performance metric. It is useful but insufficient. Typically, in an OD problem, one fixes the probability of false alarm (at say 0.01 or 0.005) or the probability of missed detection, and then computes the corresponding probability of detection or of false alarm. This needs to be done.

- The discussion of Table C3 is insufficient. When does the proposed method not beat the ‘direct’ method, and why?  What are the characteristics of those datasets?

- If the focus of the paper is on outlier detection, why are the results on OD relegated to the appendix in the supplement?

-Inference times shown in Fig 5a are meaningless without corresponding figure for accuracy. At this point, it is also not clear how FedOD is used with PCA, iForest etc. The y-axis in Fig 5d and 5e should be in log scale.

-What is the performance metric in Figure 6?

State-of-the-art
There is a large body of work on federated clustering which pays attention to data privacy. None of the following papers is cited:
1. Don Kurian Dennis, Tian Li, Virginia Smith. ICML 2021. Heterogeneity for the Win: One-Shot Federated Clustering.

2. Morris Stallmann, Anna Wilbik. Towards Federated Clustering: A Federated Fuzzy c-Means Algorithm (FFCM). On arxiv, Jan 2022.

3. Y. Wang, M. Jia, N. Gao, L. Von Krannichfeldt, M. Sun and G. Hug. Federated Clustering for Electricity Consumption Pattern Extraction. IEEE Transactions on Smart Grid, vol. 13, no. 3, pp. 2425-2439, May 2022

4. Songze Li, Sizai Hou, Baturalp Buyukates, Salman Avestimehr. Secure Federated Clustering. On arxiv May 2022.

5. W. Pedrycz. Federated FCM: Clustering Under Privacy Requirements. IEEE Transactions on Fuzzy Systems, vol. 30, no. 8, pp. 3384-3388, Aug. 2022

6. Jichan Chung, , Kangwook Lee, , Kannan Ramchandran. Federated Unsupervised Clustering with Generative Models. AAAI 2022.

There is also work on “coresets” which have been used in a federated setting for clustering (and other downstream ML tasks). For example:
7. H. Lu, M. -J. Li, T. He, S. Wang, V. Narayanan and K. S. Chan. Robust Coreset Construction for Distributed Machine Learning. IEEE Journal on Selected Areas in Communications, vol. 38, no. 10, pp. 2400-2417, Oct. 2020,

---

### Official Review · Reviewer_UDJk · 2023-10-29

**Soundness:** 2 fair
**Presentation:** 3 good
**Contribution:** 3 good
**Rating:** 6
**Confidence:** 4

**Summary:**

This paper proposes FEDOD, the first FL-based system designed for general outlier detection algorithms.
To over-come the privacy and efficiency issues of classical OD algorithms, FEDOD decomposes these algorithms into a set of basic operators and approximating their behaviors using neural networks.
FEDOD supports over several classical OD algorithms and shows great potential to be extendable to other fields like classification and clustering.
FEDOD’s accuracy and efficacy are evaluated using synthetic and benchmark datasets.

**Strengths:**

1. The challenges and approaches for designing the FL version of non-NN OD algorithms are described with clarity. Figure 1 acts as a motivating example and fig 2 is clear for describing the proposed approach.
2. The decomposition of OD algorithms -> complex operators -> simple operators,  is a clever approach and provides good modularity and extensibility.
3. providing the code of NNC for 10+ operators is a good contribution to the community.

**Weaknesses:**

1. My major concern is a Lack of theoretic explanation for the NNC, especially, how to overcome the lack of a global data view in the FL setting. It seems very difficult ' to design a loss function that allows each neural OD operator to update the model w.r.t. the local data only, to approximate the global ground truth.'
For example, Equation (1) does not always approximate the true global KNN Obj. Because a good clustering for each local agent does not mean that they are good from a global point of view. How good it is, and what are the costs by not having a global view, is an important research question. This needs more in-depth theoretic studies.

2. In Section 4.2, only showing the KNN example is not enough for readers to judge whether the approximation approach generalize to all operators. It is also important show how other common operators are approximated in the FL setting, for example the tree-based operators.

2. In equation 1, D_{i,j} is ambiguous, does it stand for D_{i,j} = cdist(X_i, X_j) ? Which violates the data preservation constraint of FL.

3. Table C3 contain the main results, it should be placed in the main text rather than in the appendix. Meanwhile, in this table, FedOD sometimes surpasses the gound truth, how is that possible. The paper should provide explanation.

**Questions:**

no

---

### Official Review · Reviewer_9wxE · 2023-10-31

**Soundness:** 2 fair
**Presentation:** 2 fair
**Contribution:** 2 fair
**Rating:** 3
**Confidence:** 4

**Summary:**

The paper proposes FEDOD for outlier detection in the federated setting. The paper considers classical outlier detection algorithms such as kNN. To enable these algorithms in the federated setting, the paper decomposes them into basic operators and then learns a neural network for each operator using FedAvg. For example, for kNN, the paper trains an NN to predict the cluster label by minimizing the intra-cluster distance and maximizing the inter-cluster distance for each local sample. Experiments show that FEDOD outperforms the baseline that each client applies outlier detection algorithm locally.

**Strengths:**

1. The research direction of considering classical outlier detection algorithms is less exploited.
2. The idea of algorithm decomposition and NN approximation for each component is interesting.

**Weaknesses:**

1. The approach misses many details. The paper only introduces how to approximate kNN with a NN. Figure 4 includes many operators. It’s not clear how to implement the operators that are not based on kNN. The paper should demonstrate more clearly how the other algorithms are implemented.

2. The approach of approximating kNN using NNs needs further clarification. I do not understand why the proposed approach works. The cluster label of different clients can be quite different and the cluster label itself does not have any meanings. The same data can be labeled as cluster 1 in client A and cluster 2 in client B. More justification on why averaging works should be provided.

3. Experiments lack important baselines. Only centralized training and individual training are compared. There are some existing approaches [1,2] that do clustering / nearest neighbor search in the federated setting.
[1] Heterogeneity for the win: One-shot federated clustering
[2] FlyNN: Fruit-fly Inspired Federated Nearest Neighbor Classification

4. The citation format is wrong. Please use \citep if the citation is not part of the sentence.

**Questions:**

1. How to enable federated OD besides kNN?

2. Why averaging NNs that predict cluster labels would help?

3. Can you add experimental results of additional baselines?

---

### Meta-Review · Area_Chair_nwcK · 2023-12-02

**Metareview:**

**Summary:**

The paper presents FEDOD, a new system designed for outlier detection (OD) in a federated learning (FL). It aims to address privacy and efficiency challenges in classical OD algorithms by decomposing these algorithms into basic operators and approximating their behaviors using neural networks. The paper claims that FEDOD supports over 20 classical OD algorithms and demonstrates its effectiveness and scalability through experiments on various datasets.

**Strengths:**

1. The paper addresses the less-explored area of applying classical OD algorithms in a federated setting.
2. The decomposition of OD algorithms into basic operators and their neural network approximation is an interesting and potentially modular approach.

**Weaknesses:**

1. There is a significant gap in the theoretical underpinning of the neural network approximations, especially in the context of lacking a global data view in FL.

2. The experimental section lacks comparisons with state-of-the-art methods in federated clustering and OD, and the choice of baselines is not adequately justified.

3. The paper is critiqued for its poor presentation and organization, making it difficult to understand the proposed method and its implications fully.

4. Several questions about the algorithm's applicability, especially regarding its extension to various types of clustering and OD problems, remain unanswered.

5. The paper does not sufficiently engage with recent and relevant literature in federated clustering and outlier detection.

**Justification For Why Not Higher Score:**

While the paper tackles an important problem and proposes an innovative approach, it falls short in theoretical explanation, methodological clarity, and comprehensive experimental validation.

**Justification For Why Not Lower Score:**

N/A

---

### Decision · Program_Chairs · 2024-01-16

Reject